# Enhancing Taxonomic Classification Consistency with Hierarchical Reasoning

## Abstract

While Vision-Language Models (VLMs) excel at visual understanding, they often fail to grasp hierarchical knowledge. This leads to common errors where VLMs misclassify coarser taxonomic levels even when correctly identifying the most specific level (leaf level). Existing approaches largely overlook this issue by failing to model hierarchical reasoning. To address this gap, we propose VL-Taxon, a two-stage, hierarchy-based reasoning framework designed to improve both leaf-level accuracy and hierarchical consistency in taxonomic classification. The first stage employs a top-down process to enhance leaf-level classification accuracy. The second stage then leverages this accurate leaf-level output to ensure consistency throughout the entire taxonomic hierarchy. Each stage is initially trained with SFT to instill taxonomy knowledge, followed by RL to refine the model's reasoning and generalization capabilities. Extensive experiments reveal a remarkable result: our VL-Taxon framework, implemented on the Qwen2.5-VL-7B model, outperforms its original 72B counterpart by over 10% in both leaf-level and hierarchical consistency accuracy on average on the iNaturalist-2021 dataset. Notably, this significant gain was achieved by fine-tuning on just a small subset of data, without relying on any examples generated by other VLMs.

## 1 Introduction

Taxonomy is a systematic classification framework that organizes objects into groups based on shared characteristics at multiple levels of granularity. A well-known example is biological taxonomy (Van Horn et al., 2021), where organisms are categorized in a hierarchical structure that typically follows the order: *Kingdom → Phylum → Class → Order → Family → Genus → Species*. Such hierarchical representations are not only fundamental for human understanding of complex relationships but also play a crucial role in visual recognition and image classification.

In recent years, large Vision-Language Models (VLMs) (Liu et al., 2023; Chen et al., 2024c; Bai et al., 2023) have been widely adopted for image classification tasks due to their strong visual understanding and question-answering capabilities. Despite these successes, recent studies (Tan et al., 2025) have highlighted that VLMs exhibit limited ability in reasoning over hierarchical structures. In particular, while these models often predict the most specific category (e.g., the *Species* level) correctly, they frequently fail to identify the correct higher-level categories. As illustrated in Figure 1 (Left), a VLM may successfully classify the *Species* of the *Heteromeles arbutifolia* (also known as *Toyon*) from the given image but still misclassifies its *Order, Family*, or *Genus*, resulting in inconsistencies across the taxonomic hierarchy. While prior work has identified this problem, no existing approach has attempted to explicitly incorporate hierarchical reasoning—an approach naturally well-suited for taxonomic classification—leaving this issue largely unresolved.

To address this problem, we begin with the intuitive hypothesis that explicitly listing the classification at each level of the taxonomy before providing the final prediction would improve overall performance. Following (Tan et al., 2025), we finetune the Qwen2.5-VL-7B-Instruct (Bai et al., 2025) model via supervised finetuning (SFT) on a small subset of the iNaturalist-Plant training set (Van Horn et al., 2021) (denoted as iNat21-Plant) and evaluate it on the test sets of both plants and animals. As shown in Table 1, we compare the original model with two approaches: (1) Default SFT, which only outputs the final answer for each level, and (2) Hierarchical SFT, which outputs predictions for all levels in sequence and then provides the final answer. We report re-

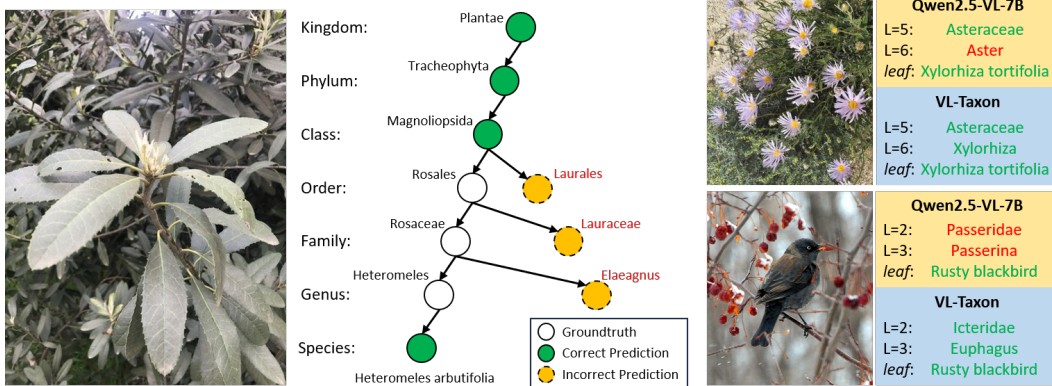

Figure 1: **Left**: Illustration of a typical plant taxonomic classification where VLMs are not able to follow the hierarchy even though their prediction at the most specific level (*leaf* level) is correct. **Right**: Examples of the predictions of the original Qwen2.5-VL-7B-Instruct and our extended VL-Taxon. $L$ denotes the level index in the test set. Correct/incorrect answers are colored green/red.

sults in terms of Hierarchical Consistent Accuracy (HCA) (Wu et al., 2024; Park et al., 2024; Tan et al., 2025)—which considers a prediction correct only if all hierarchical levels are correctly classified—and leaf-level accuracy ($\text{Acc}_{\text{leaf}}$). Additional experimental details are provided in Section 4. While Hierarchical SFT substantially improves performance on the iNat21-Plant test set, demonstrating the potential of hierarchical reasoning in taxonomic classification, we observe two major issues with this approach: (1) The listed hierarchy at different levels can be different, and (2) The generalization ability is limited.

For the first issue of inconsistent hierarchy listings, we present an illustrative example in Figure 2 (Left). For the same test image, the predicted taxonomic hierarchy differs at the *Order* and *Species* levels. To analyze this phenomenon, we summarize the Hierarchical Classification Accuracy (HCA) for each level in Figure 2 (Right), which reveals that the HCA generally improves as the classification level becomes more fine-grained. Since the benchmark (Tan et al., 2025) adopts a multiple-choice setting, we hypothesize that providing fine-grained choices within the prompt may improve HCA. To test this hypothesis, we conduct an additional experiment in which the model is trained to output the full taxonomic hierarchy given only the image, with and without the groundtruth leaf-level classification, and without any additional questions or answer choices. For fair comparison, we compute HCA only on cases where the leaf-level prediction of the unconditional (direct) listing is correct. The results in Table 2 confirm that including the leaf-level classification in the prompt improves hierarchical consistency. These findings suggest that a two-stage inference process could further enhance HCA: first, predict the most specific classification of the image; then, answer the classification question conditioned on the first-stage prediction.

Table 1: Comparison of different finetuning results

| Method | iNat21-Animal | | iNat21-Plant | |
|---|---|---|---|---|
| | HCA | $\text{Acc}_{\text{leaf}}$ | HCA | $\text{Acc}_{\text{leaf}}$ |
| Qwen2.5-VL-7B | 19.87 | 41.78 | 18.01 | 41.33 |
| Default SFT | **26.50** | **51.69** | 37.34 | 57.84 |
| Hierarchical SFT | 4.33 | 38.47 | **57.50** | **75.05** |

Table 2: HCA when the leaf level of the direct listing result is true

| Method | iNat21-Plant |
|---|---|
| Direct Listing | 79.14 |
| *Leaf* Condition | **99.52** |

For the second issue concerning generalization, Table 1 shows that hierarchical SFT tends to overfit the plant dataset due to the strong class imbalance in training. Because the model is trained exclusively on plant data, it learns to always begin its hierarchical listing with *Plantae*, leading to systematic errors such as misclassifying animals as plants when evaluated on an animal dataset. To address this limitation, we propose employing a reinforcement learning (RL) approach based on Group Relative Policy Optimization (GRPO) (Shao et al., 2024) to improve generalization while still using only the plant training dataset. GRPO is an RL algorithm designed to enhance complex reasoning in large language models (LLMs) by encouraging them to select responses that are opti-

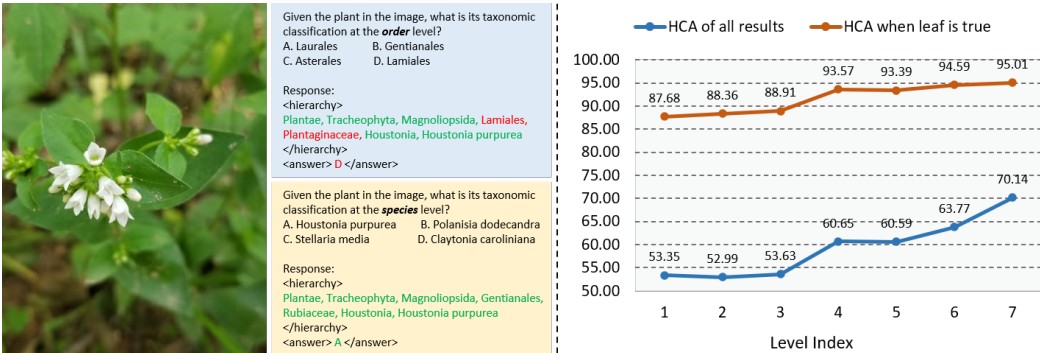

Figure 2: **Left**: Example of inconsistent hierarchical listings across different levels for the same image, with correct/incorrect answers highlighted in green/red. **Right**: HCA across different levels' listing of the hierarchy. Blue lines indicate the HCA computed over all results at a given level, whereas orange lines represent the HCA computed only for cases where the leaf-level classification listed at a certain level is correct.

mal relative to a group of candidate answers. Recent work has successfully applied GRPO to VLM finetuning (Shen et al., 2025; Yu et al., 2025a), achieving better generalization performance than naive SFT. Inspired by these findings, we adopt GRPO for our hierarchical classification setting to encourage more robust and generalizable hierarchical reasoning.

Based on the above analysis, we propose VL-Taxon, a two-stage hierarchical reasoning framework for taxonomic classification. In the first stage, the model performs a top-down reasoning process, sequentially predicting classifications from the most general to the most specific level, ultimately outputting the particular classification of the given image. In the second stage, the model conducts another top-down reasoning process conditioned on the first stage's prediction, which improves the consistency across all hierarchical levels. To further enhance generalization, we divide the training set into two disjoint subsets: one used for supervised fine-tuning (SFT) to teach the model the taxonomy knowledge, and the other for reinforcement learning with Group Relative Policy Optimization (GRPO) to improve the top-down reasoning and generalization ability. Extensive experiments demonstrate that our Qwen2.5-VL-7B-based VL-Taxon achieves up to a 30% improvement in HCA and $Acc_{leaf}$, and even surpasses its 72B-parameter counterpart across multiple benchmark datasets. Code and data will be published upon acceptance.

## 2 RELATED WORKS

**Vision Language Models** (VLMs) have been extensively studied (Zhang et al., 2024a) since the pioneering work of CLIP (Radford et al., 2021), which introduced a contrastive learning–based framework to align image-text representations and enabled zero-shot image classification. Building on this foundation, subsequent studies (Yao et al., 2021; Li et al., 2022a;b; Zhai et al., 2023; Li et al., 2023) further advanced VLMs by designing new interaction mechanisms, loss functions, and data augmentation strategies, thereby enhancing visual understanding capabilities.

Recently, large language models (LLMs) (Achiam et al., 2023) have demonstrated remarkable abilities in natural language understanding, reasoning, and generation. Leveraging these advances, large VLMs that integrate pretrained LLMs with visual encoders have achieved significantly stronger reasoning and perception compared to conventional VLMs. Notably, (Alayrac et al., 2022) pioneered the bridging of pretrained vision-only and language-only models via additional cross-attention layers, while (Liu et al., 2023) employed GPT-4 (Achiam et al., 2023) to generate multimodal training data, enabling effective alignment between vision encoders and LLMs. More recently, to strengthen the reasoning capacity of large VLMs, rule-based reinforcement learning methods, represented by GRPO (Shao et al., 2024), have been applied in the finetuning (Shen et al., 2025; Yu et al., 2025a).

Today, open-source large VLMs such as LLaVA (Liu et al., 2023; Li et al., 2024; Liu et al., 2024a;b), InternVL (Chen et al., 2024c;b;a; Zhu et al., 2025), and QwenVL (Bai et al., 2023; Wang et al., 2024;

Bai et al., 2025) have been widely adopted across a broad spectrum of vision-language tasks. Despite their impressive progress, recent work (Tan et al., 2025) has revealed that these models still struggle with hierarchical visual understanding. In particular, they often misclassify intermediate taxonomic levels, even when their predictions at the most fine-grained level are correct. This inconsistency highlights a fundamental limitation in current large VLMs, and effective solutions to address it remain largely unexplored.

**Taxonomic Hierarchical Classification** has been studied extensively for decades, even prior to the deep learning era (Marszalek & Schmid, 2007; Shahbaba & Neal, 2007; Van Horn et al., 2021; Zhao et al., 2011; Salakhutdinov et al., 2011; Deng et al., 2012; 2014). With the advent of deep learning, early hierarchical classifiers were primarily built upon convolutional neural networks (CNNs) (Yan et al., 2015; Goo et al., 2016; Ahmed et al., 2016; Zhu & Bain, 2017; Liu et al., 2018; Kim & Frahm, 2018; Chen et al., 2022). Most of these approaches adopted multi-branch architectures to capture features at different taxonomic or semantic levels for classification. Later, hierarchical visual representations were also shown to play an important role in vision transformer (ViT)-based classifiers (Dosovitskiy et al., 2020; Park et al., 2024).

With the development of CLIP (Radford et al., 2021), research focus has shifted toward VLM-based classification in a question-answering format. For example, Yi et al. (2022) proposed a hierarchical graphical knowledge representation framework for CLIP-style zero-shot image classification; Geng et al. (2023) introduced a hierarchy-aware attention mechanism to improve CLIP's classification accuracy; Wu et al. (2024) proposed a prompt-tuning method to enhance taxonomic consistency in CLIP and further introduced the Hierarchical Consistent Accuracy (HCA) metric; and Pal et al. (2024) employed hyperbolic embeddings to strengthen CLIP's hierarchical performance.

More recently, with the integration of LLMs, large VLMs have demonstrated strong performance in visual question-answering-based classification. However, recent studies reveal that these models still face notable challenges in fine-grained (Zhang et al., 2024b; Liu et al., 2024c; He et al., 2025; Yu et al., 2025b; Geigle et al., 2024; Conti et al., 2025) and hierarchical (Tan et al., 2025) classification tasks. In particular, Tan et al. (2025) established a benchmark to evaluate hierarchical consistency in taxonomic classification and highlighted that large VLMs often fail to maintain consistency across taxonomic levels. Despite these insights, prior work has not explored concrete methods to address this limitation. In this work, we fill this gap and, for the first time, show that two-stage hierarchical reasoning can substantially improve large VLMs' performance in taxonomic classification.

## 3 METHOD

Hierarchical taxonomic classification prioritizes maintaining consistency across all levels of the taxonomy rather than optimizing accuracy at a single level, which sets it apart from conventional flat classification tasks. To address the resulting challenge of ensuring coherent predictions across levels, we introduce VL-Taxon, a two-stage framework with top-down hierarchical reasoning, as illustrated in Figure 3. In the first stage, the model predicts the specific category of the given image. This predicted category is then used as a prior for the second stage to enforce consistency across all levels of the taxonomy.

### 3.1 STAGE 1: HIERARCHICAL INFERENCE FOR SPECIFIC CLASSIFICATION

In Stage 1, the model is trained to reason through the taxonomic hierarchy in a top-down manner, sequentially considering each level before predicting the most specific category of the given image. This process encourages the model to form intermediate representations that reflect hierarchical dependencies, rather than making an isolated, flat prediction. To reflect realistic deployment scenarios and prevent information leakage from fixed answer sets, the question-answering in this stage is formulated in an open-set manner, where the model must generate the correct category name rather than select from a predefined list.

As discussed in the Introduction, explicitly reasoning through the hierarchy has been shown to improve both prediction accuracy and consistency, as it forces the model to check each intermediate level before committing to the final answer. Therefore, as illustrated in Figure 3, we explicitly instruct the model to perform top-down reasoning, from the general level to the specific level. The

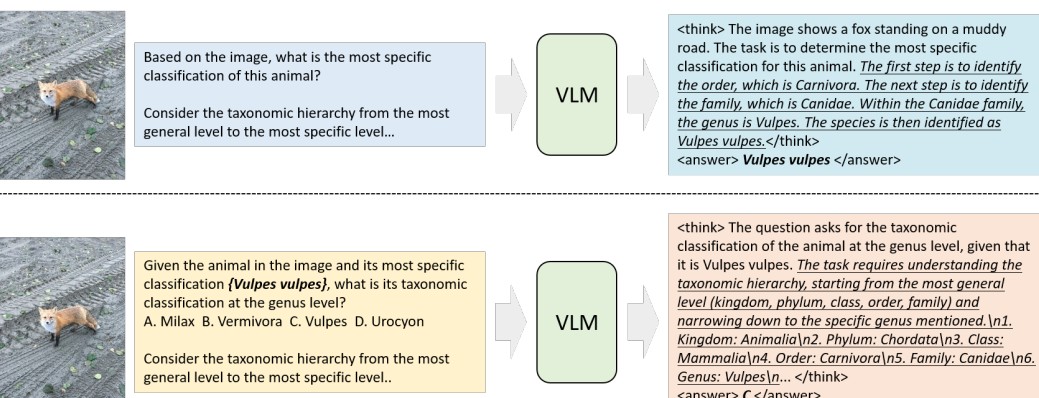

Stage 1: Specifically classify of the image

Stage 2: Answer the question based on Stage 1's output

Figure 3: Framework for the proposed VL-Taxon with two-stage hierarchical reasoning. **Stage 1**: Output the specific classification of the given image based on taxonomic hierarchical reasoning. **Stage 2**: Answer the specific question based on Stage 1's output to align the taxonomic hierarchy. The taxonomic hierarchical reasoning part in the thinking process is underlined in the example.

final output is the predicted name of the most specific category, which will then be used as a prior for Stage 2.

### 3.2 STAGE 2: QUESTION ANSWERING BASED ON THE SPECIFIC CLASSIFICATION

In Stage 2, the model leverages the specific prediction obtained in Stage 1 to answer classification questions under various evaluation settings. This stage is designed to enforce hierarchical consistency across all levels of the taxonomy. As discussed previously, conditioning the model on the most specific classification encourages its intermediate predictions to remain logically coherent with the leaf-level result, thereby reducing contradictions within the hierarchy.

As illustrated in Figure 3, we explicitly prompt the model to perform a second top-down reasoning pass, now conditioned on the Stage 1 prediction. This guided reasoning process encourages the model to refine its earlier decisions, leading to improved accuracy for each hierarchical level and better overall consistency. Together, Stages 1 and 2 form a closed-loop reasoning pipeline that combines fine-grained recognition with globally coherent taxonomy predictions.

### 3.3 FINETUNING WITH GROUP RELATIVE POLICY OPTIMIZATION

As analyzed in the Introduction, SFT is limited in its ability to generalize either factual knowledge or reasoning patterns to previously unseen categories. To address this limitation, we employ Group Relative Policy Optimization (GRPO) (Shao et al., 2024; Guo et al., 2025), a recently proposed rule-based reinforcement learning (RL) algorithm, to enhance the model's generalization ability for hierarchical reasoning on unseen datasets. Unlike conventional RL approaches such as (Schulman et al., 2017; Ouyang et al., 2022; Rafailov et al., 2023), which rely on human feedback to evaluate the policy, GRPO optimizes the policy by comparing the relative rewards within a group of candidate responses $o_1, o_2, \ldots, o_G$ to the same input question $q$. This relative-comparison paradigm eliminates the need for expensive human annotations, making it suitable for this hierarchical classification.

Formally, the optimization objective of GRPO for a policy $\pi_\theta$ can be expressed as

$$\mathcal{J}_{\text{GRPO}}(\theta) = \mathbb{E}_{[q \sim P(Q), \{o_i\}_{i=1}^G \sim \pi_{\theta_{\text{old}}}(O|q)]}$$

$$\frac{1}{G} \sum_{i=1}^{G} \left\{ \min\left[ \frac{\pi_\theta(o_i|q)}{\pi_{\theta_{\text{old}}}(o_i|q)} A_i, \text{clip}\left( \frac{\pi_\theta(o_i|q)}{\pi_{\theta_{\text{old}}}(o_i|q)}, 1 - \epsilon, 1 + \epsilon \right) A_i \right] - \beta \mathbb{D}_{KL}(\pi_\theta || \pi_{\text{ref}}) \right\} \quad (1)$$

where $A_i$ is the advantage, $\epsilon$ and $\beta$ are hyperparameters. Given the reward $r_i$ for each $o_i$, the advantage $A_i$ is estimated by:

$$A_i = r_i - \frac{\text{mean}\{r_1, r_2, \ldots, r_N\}}{\text{std}\{r_1, r_2, \ldots, r_N\}} \tag{2}$$

In this work, we employ two types of rewards: format reward and accuracy reward, each taking a binary value of 1 if satisfied and 0 otherwise. The format reward verifies whether the model's output strictly follows the expected response structure, i.e., *<think>...</think> <answer>...</answer>*. The accuracy reward measures whether the model's prediction matches the groundtruth. For Stage 1 training samples, the reward is granted only when the predicted category name exactly matches the ground-truth label, thereby encouraging precise predictions at the most specific taxonomic level. For Stage 2 training samples, the reward is assigned if the predicted answer letter is correct, which aligns with the multiple-choice formulation used in (Tan et al., 2025).

## 4 EXPERIMENTS

### 4.1 EXPERIMENTAL SETUP

#### 4.1.1 DATASET

For the training dataset, we follow (Tan et al., 2025) and adopt a subset of the iNat21-Plant (Van Horn et al., 2021) training split. Each question is formulated as a four-option multiple-choice query with a single correct answer. We use 3,771 out of 4,271 species for training, sampling 10 images per species. For evaluation, we conduct experiments on the benchmark introduced in (Tan et al., 2025), using the test sets of iNat21-Animal (Van Horn et al., 2021), iNat21-Plant (Van Horn et al., 2021), and CUB-200 (Wah et al., 2011). All evaluations are performed under the similar-choice protocol, where the distractor options in the test sets are selected based on the labels with the highest cosine similarity scores between the text labels and image embeddings, computed using SigLIP (Zhai et al., 2023). This design not only increases the difficulty of the task by introducing visually and semantically similar alternatives but also makes the evaluation more realistic for practical applications.

It is important to note that we exclude the datasets derived from ImageNet (Deng et al., 2009) in (Tan et al., 2025) for two primary reasons. First, although their hierarchies are constructed from WordNet (Miller, 1995), the names and definitions of each taxonomic level are not provided, limiting their interpretability and practical value. Second, these datasets contain a large number of ambiguous choices with overlapping semantic scopes, partly due to the lack of level definitions and the absence of human validation after automated distractor selection based on SigLIP. For example, a question may present options such as *machine, electronic device, electronic equipment* for an image of a *desktop computer*, with the groundtruth answer being *machine*, which is arguably ambiguous.

#### 4.1.2 TRAINING CONFIGURATION

Our experiments are conducted using QwenVL-2.5-7B-Instruct (Bai et al., 2025) as the backbone model. To efficiently utilize the training data, we partition the training set into two equal subsets by species. The model is first fine-tuned on the first subset using supervised fine-tuning (SFT) to acquire fundamental knowledge for taxonomic classification. Subsequently, we apply GRPO-based reinforcement learning (RL) on the second subset to enhance the model's hierarchical reasoning capabilities and improve its generalization to unseen categories.

We employ LoRA (Hu et al., 2022) to finetune the model on each subset for one epoch. The global batch size is 128, and the learning rate is $5 \times 10^{-5}$. The LoRA rank and $\alpha$ are 64. For GRPO, group size $G$ is 8, and the $\beta$ parameter for the KL-divergence is 0.4.

#### 4.1.3 EVALUATION METRICS

Following the previous works (Wu et al., 2024; Tan et al., 2025), we primarily focus on the evaluation on the hierarchical consistent accuracy. We also test the leaf-level accuracy, which can be regarded as the upper bound of hierarchical consistent accuracy.

**Hierarchical Consistent Accuracy (HCA)** strictly requires the classification across all taxonomic levels to be true, which is defined as:

$$\text{HCA} = \frac{1}{N} \sum_{i=1}^{N} \prod_{j=1}^{L^i} \mathbb{1}[f_\theta(x^i; \mathcal{Y}_j) = y_j^i] \tag{3}$$

where $i$ denotes the index of image, $L^i$ denotes the depth of the taxonomic hierarchy of the $i$-th image. $\mathbb{1}$ is the indicator function. $f_\theta$ is the classifier function. $x_i$ is the input image and prompt. $\mathcal{Y}_j$ and $y_j^i$ are the candidate label set and groundtruth label at the $j$-th level of the $i$-th image.

**Leaf-level accuracy (Acc$_{\text{leaf}}$)** measures the prediction correctness at leaf-level, defined as:

$$\text{Acc}_{\text{leaf}} = \frac{1}{N} \sum_{i=1}^{N} \mathbb{1}[f_\theta(x^i; \mathcal{Y}_L) = y_L^i] \tag{4}$$

By comparing the definitions of HCA and Acc$_{\text{leaf}}$, we observe that a prediction can only be counted as correct under HCA if its leaf-level prediction is also correct. Consequently, Acc$_{\text{leaf}}$ serves as an upper bound for the HCA metric.

### 4.2 QUANTITATIVE RESULT

In Table 3, we compare our VL-Taxon with six state-of-the-art open-source large VLMs: LLaVA-OV-7B (Li et al., 2024), InternVL2.5-8B (Chen et al., 2024a), InternVL3-8B Zhu et al. (2025), and Qwen2.5-VL in its 7B, 32B, and 72B-Instruct variants (Bai et al., 2025). The results of the compared baseline methods are from (Tan et al., 2025). VL-Taxon consistently outperforms all baseline models, including the substantially larger Qwen2.5-VL-72B-Instruct, on both the iNat21-Animal and iNat21-Plant datasets, while achieving competitive performance on CUB-200. Remarkably, when compared to its backbone model, Qwen2.5-VL-7B-Instruct, VL-Taxon achieves up to a 45% relative improvement in HCA on the iNat21-Plant dataset, demonstrating the effectiveness of the proposed two-stage hierarchical reasoning framework. This improvement is particularly notable given that VL-Taxon utilizes the same model capacity as its backbone, highlighting that the performance gain stems from the improved training and inference strategy rather than increased model size.

Moreover, VL-Taxon is fine-tuned exclusively on a subset of plant data and is never exposed to animal examples during finetuning. Despite this, it achieves state-of-the-art performance on the iNat21-Animal dataset, providing strong evidence of its ability to generalize beyond the training domain. The strong cross-domain generalization suggests that VL-Taxon is capable of learning transferable hierarchical reasoning patterns that are not limited to a single taxonomic group. These results collectively highlight not only the superior performance of VL-Taxon in hierarchical taxonomic classification but also its robustness for broader applications involving unseen domains.

### 4.3 ABLATION STUDY

#### 4.3.1 TWO-STAGE HIERARCHICAL REASONING

Table 4 compares VL-Taxon with its ablated variants, where either the hierarchical reasoning process or the two-stage configuration is removed during inference. The results clearly show that omitting either component leads to a substantial drop in HCA. The top-down hierarchical reasoning process is particularly crucial: it first predicts the higher-level taxonomic categories—which are generally easier to classify—and then progressively refines predictions at the more specific levels, thereby improving their accuracy. Interestingly, Acc$_{\text{leaf}}$ on the iNat21-Animal dataset is slightly higher without reasoning. This may be attributed to the original Qwen2.5-VL-7B model already encoding knowledge of animal species, which allows correct leaf-level predictions without explicit reasoning, further reinforced by our training. Nevertheless, despite the marginally higher Acc$_{\text{leaf}}$, the HCA is lower without reasoning, underscoring the importance of hierarchical reasoning in maintaining consistency across intermediate levels—the core challenge of this taxonomic classification task.

Regarding the two-stage framework, when the model predicts intermediate levels directly without conditioning on the leaf-level prediction from Stage 1, the HCA decreases substantially. This indicates that the leaf-level prediction serves as an essential prior, guiding intermediate-level predictions and ensuring coherence across the entire taxonomy. These findings are consistent with the

Table 3: Comparison of HCA and $\text{Acc}_{\text{leaf}}$ of open-source large vision language models. The best scores are bolded, and the second-best scores are underlined.

| Method | iNat21-Animal | | iNat21-Plant | | CUB-200 | |
|---|---|---|---|---|---|---|
| | HCA | $\text{Acc}_{\text{leaf}}$ | HCA | $\text{Acc}_{\text{leaf}}$ | HCA | $\text{Acc}_{\text{leaf}}$ |
| LLaVA-OV-7B | 4.53 | 26.47 | 4.46 | 27.51 | 11.51 | 44.23 |
| InternVL2.5-8B | 8.52 | 27.65 | 5.56 | 28.36 | 22.07 | 45.56 |
| InternVL3-8B | 11.93 | 35.40 | 8.68 | 36.39 | 25.75 | 50.52 |
| Qwen2.5-VL-7B | 19.43 | 41.33 | 17.67 | 41.61 | 43.76 | 65.50 |
| Qwen2.5-VL-32B | 26.90 | 46.98 | 24.64 | 48.57 | 56.80 | 69.00 |
| Qwen2.5-VL-72B | 35.73 | 54.20 | 32.82 | 55.00 | **66.36** | **75.04** |
| VL-Taxon (Ours, 7B) | **43.73** | **60.72** | **63.04** | **74.36** | 60.67 | 70.92 |

Table 4: Comparison of the reasoning and two-stage configurations.

| Methods | iNat21-Animal | | iNat21-Plant | | CUB-200 | |
|---|---|---|---|---|---|---|
| | HCA | $\text{Acc}_{\text{leaf}}$ | HCA | $\text{Acc}_{\text{leaf}}$ | HCA | $\text{Acc}_{\text{leaf}}$ |
| w/o reasoning | 41.02 | **62.05** | 58.12 | 72.42 | 55.32 | 69.07 |
| w/o first stage | 34.63 | 58.34 | 49.56 | 70.33 | 54.56 | **71.14** |
| VL-Taxon | **43.73** | 60.72 | **63.04** | **74.36** | **60.67** | 70.92 |

pilot experiments presented in the Introduction and further validate the design choices underlying our two-stage hierarchical reasoning framework.

### 4.3.2 INTERMEDIATE LEVELS' CONSISTENCY

As discussed in previous works (Tan et al., 2025; Wu et al., 2024) and reiterated in the Introduction, existing VLMs often achieve correct classification at the leaf level but tend to make errors in the intermediate levels, resulting in inconsistencies across the taxonomic hierarchy. In earlier experiments, we have shown that VL-Taxon improves both HCA and $\text{Acc}_{\text{leaf}}$ compared with its backbone Qwen2.5-VL-7B across all datasets. However, it remains unclear whether these HCA improvements are primarily due to gains at the leaf level or from enhanced consistency at the intermediate levels. To address this question, we conduct a more detailed investigation of VL-Taxon's impact on intermediate-level predictions.

Specifically, we compare VL-Taxon with the original Qwen2.5-VL-7B-Instruct and its default supervised finetuned variant, denoted as Qwen2.5-VL-7B-SFT, which is trained by directly answering each question in the training set. To isolate the contribution of intermediate levels, we compute the HCA conditioned on the correctness of the leaf-level prediction, denoted as HCA (L), as shown in Table 5. This setup controls for leaf-level performance and provides a clearer measure of hierarchical consistency. The results reveal that VL-Taxon achieves substantial improvements in HCA (L), demonstrating that the observed gains in overall HCA are not solely attributable to leaf-level accuracy, but also stem from stronger consistency across intermediate levels. Thus, VL-Taxon effectively addresses the hierarchical inconsistency issues highlighted in prior work.

Moreover, although Qwen2.5-VL-7B-SFT achieves modest improvements in HCA (L) on the iNat21-Plant dataset and consistently enhances Accleaf across datasets, its improvements in HCA (L) on other datasets are limited. In contrast, VL-Taxon significantly improves both HCA (L) and Accleaf across all datasets, confirming its effectiveness and demonstrating its strong generalization ability across domains and taxonomic structures.

Figure 4 further illustrates this advantage by comparing the classification accuracy of Qwen2.5-VL-7B, Qwen2.5-VL-7B-SFT, and VL-Taxon at each taxonomic level on the iNat21-Plant and CUB-200 datasets. With the exception of the top three levels in iNat21-Plant—which are relatively easy to classify and therefore exhibit smaller differences—VL-Taxon consistently and significantly outperforms both baselines across intermediate and fine-grained levels. These results confirm that VL-Taxon not only boosts leaf-level accuracy but also substantially enhances intermediate-level classification accuracy and hierarchical consistency, thereby achieving more reliable and biologically coherent taxonomic predictions.

Table 5: Comparison of the intermediate levels' hierarchical consistent accuracy when the leaf-level prediction is correct (HCA (L)) and the leaf-level accuracy ($Acc_{leaf}$).

| Methods | iNat21-Animal | | iNat21-Plant | | CUB-200 | |
|---|---|---|---|---|---|---|
| | HCA (L) | $Acc_{leaf}$ | HCA (L) | $Acc_{leaf}$ | HCA (L) | $Acc_{leaf}$ |
| Qwen2.5-VL-7B | 47.57 | 41.78 | 43.58 | 41.33 | 66.21 | 64.72 |
| Qwen2.5-VL-7B-SFT | 51.27 | 51.69 | 64.56 | 57.84 | 67.20 | 68.61 |
| VL-Taxon | **72.01** | **60.72** | **84.78** | **74.36** | **85.54** | **70.92** |

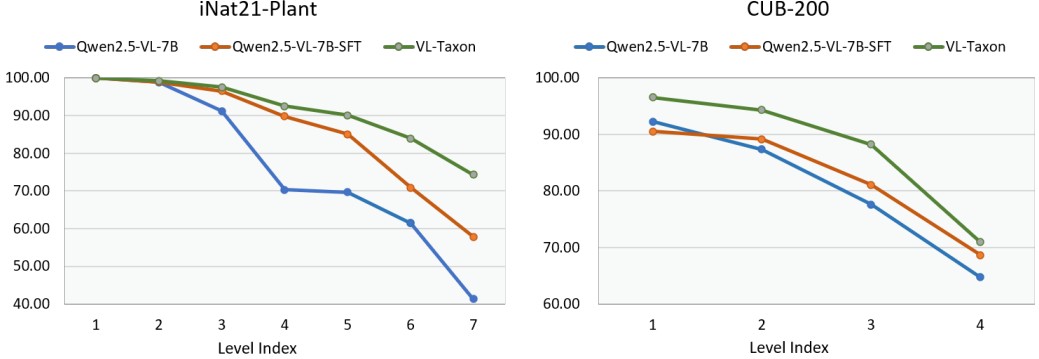

Figure 4: Comparison of classification accuracy at each level on iNat21-Plant (L) and CUB-200 (R).

Table 6: Comparison between direct GRPO finetuning on the full dataset and our hybrid approach.

| Methods | iNat21-Animal | | | iNat21-Plant | | | CUB-200 | | |
|---|---|---|---|---|---|---|---|---|---|
| | HCA | $Acc_{leaf}$ | TKs↓ | HCA | $Acc_{leaf}$ | TKs↓ | HCA | $Acc_{leaf}$ | TKs↓ |
| Full GRPO | 32.61 | 53.83 | 160.08 | 48.00 | 64.36 | 159.25 | 54.71 | 69.09 | 156.46 |
| Hybrid | **43.73** | **60.72** | **92.92** | **63.04** | **74.36** | **93.94** | **60.67** | **70.92** | **93.57** |

### 4.3.3 GRPO FINETUNING CONFIGURATION

Table 6 presents a comparison between two training strategies: (1) direct fine-tuning with GRPO on the entire training dataset, and (2) our hybrid strategy, where the dataset is evenly split, with the first half used for SFT and the second half for GRPO. For both cases, we employ the two-stage reasoning framework to ensure a fair comparison. In addition to HCA and $Acc_{leaf}$, we also evaluate the average number of tokens generated during prediction, denoted as TKs. Since TKs directly influence both GRPO finetuning and inference efficiency, a relatively lower TK count is generally better when the reasoning process and the result are reasonable. The results indicate that, without the prior taxonomic knowledge introduced by SFT, directly applying GRPO on the full dataset results in longer reasoning chains yet yields limited accuracy gains. In contrast, our hybrid training configuration not only achieves superior accuracy but also produces more efficient reasoning, further demonstrating its effectiveness and efficiency.

## 5 CONCLUSION

In this paper, we address the challenge of improving consistency in taxonomic hierarchical classification with VLMs. To this end, we introduce VL-Taxon, a two-stage hierarchy-aware reasoning framework combined with a hybrid training strategy. Specifically, VL-Taxon is first finetuned on half of the dataset via SFT to acquire foundational taxonomic knowledge, and subsequently optimized with GRPO on the remaining half to enhance hierarchical reasoning and generalization. Extensive experiments demonstrate that our Qwen2.5-VL-7B-based model, trained on a relatively small and domain-limited subset, substantially improves hierarchical taxonomic classification. Remarkably, VL-Taxon elevates the performance of the 7B model to a level competitive with its 72B counterpart across diverse domains, underscoring both the efficiency and scalability of the proposed approach.

## ETHICS STATEMENT

This work complies with the ICLR Code of Ethics. All experiments are conducted on publicly available datasets in accordance with their respective licenses. No additional human or animal subjects are involved. The proposed method is intended solely for research purposes and does not pose any foreseeable risks to security or privacy.

## REPRODUCIBILITY STATEMENT

For reproducibility, we provide a comprehensive description of the proposed method in Section 3. Section 4 presents detailed information about the experimental setup, including models, datasets, and training configurations. To ensure that all reported results can be reliably reproduced, we will release the source code and data upon acceptance.

## USAGE OF LARGE LANGUAGE MODELS

For the preparation of this paper, Large Language Models (LLMs) were used solely as writing aids for polishing text. They were not used for retrieval, discovery, or research ideation. Beyond writing assistance, since this work focuses on large Vision-Language Models (VLMs) built upon LLMs, our experiments naturally involve running evaluations and training on LLM-based architectures.

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
