# OpenReview forum: "Enhancing Taxonomic Classification Consistency with Hierarchical Reasoning"
_ICLR.cc/2026/Conference — ICLR 2026 Conference Withdrawn Submission_

### Official Review · Reviewer_KRzm · 2025-10-29

**Soundness:** 2
**Presentation:** 2
**Contribution:** 2
**Rating:** 4
**Confidence:** 3

**Summary:**

The paper discusses that while VLMs perform well in visual understanding, they have a problem in grasping hierarchical relations and classifying coarser classes, which can result in bad hierarchical reasoning. The paper proposes VL-Taxon, a two-stage hierarchy-based reasoning framework, to improve leaf-level and higher-level classification accuracy. In the first stage, the model generates classes from the finest level to the coarsest level. In the second stage, the reasoning is performed conditioning on the first stage prediction.

**Strengths:**

The paper is well-motivated. The paper points out a valid question and challenge: lack of hierarchical consistency in VLMs.

The writing is without problems and well-structured.

The idea is intuitive and simple. The experiments show that the idea works well.

The paper compares the results with the Qwen2.5-VL and shows improvements over the model. The paper provides comparisons on three datasets and shows improvements.

**Weaknesses:**

The paper uses reinforcement learning; however, in the writing, it is not clear what the pipeline is. Also, the paper discusses SFT and hierarchical SFT.

The experimental setup for Table 1 is not clear. How is the default SFT and hierarchical performed?
In Table 2, comparing leaf condition and direct listing, what if the model is memorizing the hierarchy and overfitting?
Tan et al 2025 includes results using LoRA, where the results improve (Tables 4 and 5). I wonder why these results are not included.
Why is the paper not including GPT as Tan et al 2025 does?
I wonder if the paper can compare the idea with protect paper and compare the performance and show how the ideas are different.

Equation 2 is wrong.
The abstract mentions SFT when the paper has not introduced what SFT means yet.
Lines 421 and 423, acc leaf is written wrong.

**Questions:**

Please check the weaknesses.

---

### Official Review · Reviewer_CRLF · 2025-10-31

**Soundness:** 3
**Presentation:** 3
**Contribution:** 2
**Rating:** 4
**Confidence:** 4

**Summary:**

The paper tackles the important problem of hierarchical inconsistency in large vision-language models (VLMs), proposing a two-stage reasoning framework (VL-Taxon) that first predicts fine-grained (leaf-level) categories and then refines the full taxonomy tree for consistency. The model is trained with supervised fine-tuning and GRPO-based reinforcement optimization. Experiments on iNaturalist and CUB show notable improvements over existing large-scale VLMs.

**Strengths:**

**1. Timely and meaningful problem.** The paper addresses a real and underexplored weakness of current large-scale VLMs—hierarchical inconsistency. This direction is important and worth continued attention.


**2. Empirical effectiveness**. The proposed method achieves superior performance compared to other VLM baselines on fine-grained taxonomic datasets, suggesting that hierarchical reasoning can indeed improve structured understanding.

**Weaknesses:**

**1. Questionable use of the VQA setting.**
 While the paper follows a VQA-style format, it’s not clear that this setup is necessary. Since free-form text outputs can naturally represent hierarchical answers, it’s unclear what is gained by converting the task into VQA format. Other works (e.g., H-CAST [1]) demonstrate inconsistency directly through classification without such framing.


**2. Unclear justification for two-stage design.**
 The method predicts the most specific (leaf) node and then refines coarse predictions based on it. However, given a predefined taxonomy, all parent nodes can be deterministically derived from the leaf. It’s not fully explained why explicit reasoning is needed instead of simple taxonomy lookup.


**3. Weak motivation for using GRPO.**
 The choice of GRPO for enhancing generalization is insufficiently justified. It seems more like adopting a recent RL technique rather than one particularly suited to hierarchical reasoning. A stronger rationale or comparative study with alternative methods would help.


**4. Limited analysis and interpretability.**
 The ablation studies focus solely on numerical performance changes within benchmarks. More qualitative analyses—such as visual examples of where inconsistency occurs or how the method corrects specific cases—would significantly strengthen the paper’s clarity and insight.

**Questions:**

Please refer to the weaknesses above. In particular, I would appreciate clarification on:

1. The necessity of using a VQA-style formulation instead of direct classification. Can this method be applied to hierarchical classification tasks as in [1]?

2. The motivation for predicting coarse labels from the leaf node rather than using predefined taxonomy relations.

3. The specific reason for adopting GRPO over other possible reinforcement or alignment methods.

4. Any qualitative or interpretive evidence that demonstrates how the proposed reasoning actually improves hierarchical consistency.

[1] Visually Consistent Hierarchical Image Classification, ICLR, 2025

---

### Official Review · Reviewer_WqLj · 2025-11-01

**Soundness:** 3
**Presentation:** 3
**Contribution:** 3
**Rating:** 4
**Confidence:** 3

**Summary:**

The paper introduces VL-Taxon, a two-stage, hierarchy-aware reasoning framework to improve both leaf-level accuracy and Hierarchical Consistent Accuracy (HCA) in taxonomic classification with VLMs. Stage-1 performs explicit top-down reasoning to predict the most specific (leaf) class in an open-set manner; Stage-2 answers downstream hierarchical questions conditioned on the Stage-1 leaf prediction to enforce cross-level consistency. Training uses a hybrid SFT→GRPO recipe (LoRA, one epoch each, group size 8) on a subset of iNat21-Plant; evaluation spans iNat21-Plant/Animal and CUB-200 under a similar-choice protocol. Empirically, a 7B Qwen2.5-VL backbone trained with VL-Taxon outperforms the 72B version on HCA and leaf accuracy on iNat21 datasets and shows large gains vs. LLaVA-OV and InternVL baselines; ablations suggest both the two-stage design and explicit reasoning are necessary.

**Strengths:**

Directly addresses the widely observed hierarchy failure mode in VLMs with an inference + training recipe rather than only prompting.

Stage-1 open-set leaf prediction reduces answer-set leakage; Stage-2 conditioning formalizes hierarchy alignment; GRPO rewards reflect both formatting and correctness.

7B surpassing 72B (Qwen2.5-VL) highlights training/inference strategy over sheer size—consistent with broader open-source reports on Qwen2.5-VL capability.

Hybrid SFT→GRPO aligns with the trajectory of RL for reasoning in LLMs/VLMs.

**Weaknesses:**

The paper compares mainly to general VLMs. It should include ProTeCt (HCA/MTA metrics origin) and hierarchy-aware CLIP variants (hyperbolic embeddings, hierarchical attention) under the same similar-choice protocol to isolate the benefit of two-stage reasoning vs. better text encoders/prompts.

Finetuning exclusively on plants may bias reasoning templates (authors note Plantae priors). The cross-domain gains are impressive but would be stronger with balanced plant/animal SFT or with unseen-taxonomy splits and leakage checks.

Results rely on a multiple-choice “similar-choice” setup (SigLIP similarity). Open-ended taxon name generation and noisy real-world labels could behave differently. An open-ended, non-MC evaluation and calibration of hierarchical uncertainty would improve external validity.

While LoRA ranks, lr, batch sizes, and GRPO G/β are given, training token counts, wall-clock, GPU types, and prompt templates (full) would help reproducibility and fair cost comparisons against single-stage SFT.

Include newer LLaVA-OV 1.5 releases and InternVL3.5 variants measured under the same hierarchy benchmark for completeness.

**Questions:**

Please add ProTeCt and hyperbolic CLIP baselines under your protocol (same images, options), and report HCA/MTA side-by-side.

What is the effect of removing format reward, or using partial-credit hierarchical rewards (e.g., +1 for each correct node on the path) vs. 0/1?

How does Stage-1 behave with unseen genus/species strings (misspellings, synonyms)? Any normalization/canonicalization step?

---

### Official Review · Reviewer_2YhU · 2025-11-01

**Soundness:** 2
**Presentation:** 3
**Contribution:** 2
**Rating:** 6
**Confidence:** 3

**Summary:**

The paper proposes VL-Taxon, a two-stage top-down hierarchical reasoning framework (Stage 1: predict leaf → Stage 2: answer all levels conditioned on Stage 1) plus GRPO-based hybrid fine-tuning (SFT→GRPO). It reports improvements in HCA and Acc_leaf on iNat21-Plant/Animal, CUB-200, claiming the 7B model surpasses 72B on some sets (Tables 3, 4-5-6 and §4). However, ImageNet-derived hierarchical benchmarks are excluded, and results are presented with single seed, no standard deviations, making generalization and reproducibility assessment difficult.

**Strengths:**

1. Appropriate problem definition: Explicitly distinguishes HCA (Eq. 3) from leaf accuracy (Eq. 4), clarifying the HCA ≤ Acc_leaf relationship. Proper targeting of hierarchical consistency.
2. Simple procedure with consistent improvement signals: Two-stage reasoning and hybrid training (Tables 4, 6) show broad HCA gains. The HCA(L) analysis (leaf-correct conditional HCA, Table 5) separating intermediate-level consistency improvements is compelling.

**Weaknesses:**

1. Lack of statistical rigor (single seed, no standard deviations)
All numbers in Tables 3-6 are single values without std/CI. GRPO and prompt-based pipelines are sensitive to seed/option construction, making current one-shot numbers insufficient for reproducibility/significance assessment. Require ≥3-5 seed mean±std and significance tests.
2. Unanalyzed cascade failure modes
Since Stage 2 is conditioned on Stage 1 leaf result, Stage 1 errors could lead Stage 2 to solidify consistent but incorrect hierarchies. Authors analyzed success cases with HCA(L) (Table 5) but not leaf-incorrect conditional failure patterns (e.g., "Does Stage 2 mitigate/amplify when leaf is wrong?"). Essential for deployment stability assessment.

**Questions:**

Major questions
1. Seed/variance: Can you provide seed≥3 results with statistical significance for Tables 3-6?
2. Choice bias: Can you compare similar-choice impact (human-verified vs SigLIP choices)? Open-ended performance?
3. Cascade analysis: Quantitative impact of Stage 2 on HCA when leaf is incorrect (mitigation/amplification rates)?

---

### Official Review · Reviewer_LeDB · 2025-11-02

**Soundness:** 2
**Presentation:** 1
**Contribution:** 2
**Rating:** 2
**Confidence:** 4

**Summary:**

The paper tackles a well-observed failure mode in vision–language models (VLMs): even when leaf (species) predictions are correct, intermediate taxonomic levels are inconsistent. It proposes VL‑Taxon, a two‑stage, top‑down hierarchical reasoning framework: Stage 1 performs open‑set specific (leaf) classification by explicitly reasoning through the hierarchy; Stage 2 conditions on the Stage 1 leaf to answer multiple‑choice questions at each level, enforcing cross‑level consistency. A hybrid first SFT then GRPO regime (dataset split by species) further stabilizes formatting and accuracy. Experiments on iNat21‑Animal/Plant and CUB‑200 show sizable gains in hierarchical consistent accuracy (HCA) and leaf accuracy over strong VLM baselines.

**Strengths:**

1. Ablations diagnose what matters. Removing top‑down reasoning or the first stage degrades HCA across datasets, and HCA(L) analyses confirm gains arise from better intermediate‑level coherence, not only leaves.
2. Cross‑domain generalization despite one‑domain fine-tuning. Trained solely on iNat21‑Plant, the model achieves SOTA on iNat21‑Animal, underscoring transferability of the hierarchical procedure.

**Weaknesses:**

1. Motivation is weak in the Introduction. The paper does not crisply state the concrete research question upfront (what, precisely, is broken and why existing approaches fail), nor does it articulate a falsifiable hypothesis that the later sections test. This blurs the problem–solution–evidence thread.

2. Problem statements are imprecise and under-specified. The two stated issues—(i) inconsistent hierarchical listings across levels and (ii) limited generalization—are framed as symptoms rather than formally defined problems. There is no clear operational definition, scope (which taxonomic depths, which domains), or measurable target for “generalization,” making it hard to judge whether the method addresses them.

3. Missing simple supervised CNN/ViT baselines and rationale for using VLMs. A small supervised CNN/ViT trained on the same data can achieve higher leaf accuracy (and thus HCA via ancestor mapping). The paper neither reports such baselines nor justifies the added complexity and cost of VLMs with commensurate benefits (e.g., open-worldness, label sparsity, or zero-shot transfer).

4. Undefined term: “unconditional (direct) listing.” The phrase is introduced (lines 89–90) without a prior, explicit definition or a precise contrast with “conditional” listing. It is unclear whether it refers to free-form hierarchical generation, MCQ without priors, or another setting.

5. Ambiguous visuals: unclear boundaries in Fig. 1 (Left) and Fig. 2 (Left). The figures lack clear delineation (boxes/edges/level separators), making it difficult to see where one level/path ends and another begins. This weakens the diagnostic message those figures are supposed to convey.

6. Tables 1 and 2 are insufficiently motivated and prematurely placed. The paper does not state the research questions first and then use the tables to test specific hypotheses. As presented, it is unclear what each table is designed to prove (e.g., which hypothesis about listings, priors, or generalization), why these results appear before a rigorous problem statement, and how they tie into the subsequent method.

7. Limited technical contribution. The evaluations are largely reuses of existing pieces: the training objectives in Eq. (1)–(2) follow standard likelihood/reward formulations, and the metrics in Eq. (3)–(4) (hierarchical consistency and leaf accuracy) are definitions rather than new measures. The proposed two-stage pipeline mainly rearranges known components (prompted hierarchical listing + conditioned MCQ + off-the-shelf GRPO/LoRA) without introducing a new model architecture, loss, or theory. As a result, the contribution feels procedural/engineering rather than technically novel, and the paper would benefit from clearer articulation of what is genuinely new versus what is borrowed.

**Questions:**

See the weaknesses.

---

### Note · Authors · 2025-11-14

I have read and agree with the venue's withdrawal policy on behalf of myself and my co-authors.